# Literacy overrides effects of animacy: A picture-naming study with pre-literate German children and adult speakers of German and Arabic

Sarah Dolscheid[1‡]*, Judith Schlenter[2‡], Martina Penke[1]

1 Department of Rehabilitation and Special Education, University of Cologne, Cologne, Germany,
2 Department of Language and Culture, UIT The Arctic University of Norway, Tromsø, Norway

‡ SD and JS contributed equally to this work and shared first authorship on this work.
* sarah_dolscheid@gmx.de

**Data Availability Statement:** The data files and the R code for all models are openly available on OSF

## Abstract

Animacy plays a key role for human cognition, which is also reflected in the way humans process language. However, while experiments on sentence processing show reliable effects of animacy on word order and grammatical function assignment, effects of animacy on conjoined noun phrases (e.g., *fish and shoe* vs. *shoe and fish*) have yielded inconsistent results. In the present study, we tested the possibility that effects of animacy are outranked by reading and writing habits. We examined adult speakers of German (left-to-right script) and speakers of Arabic (right-to-left script), as well as German preschool children who do not yet know how to read and write. Participants were tested in a picture naming task that presented an animate and an inanimate entity next to one another. On half of the trials, the animate entity was located on the left and, on the other half, it was located on the right side of the screen. We found that adult German and Arabic speakers differed in their order of naming. Whereas German speakers were much more likely to mention the animate entity first when it was presented on the left than on the right, a reverse tendency was observed for speakers of Arabic. Thus, in literate adults, the ordering of conjoined noun phrases was influenced by reading and writing habits rather than by the animacy status of an entity. By contrast, pre-literate children preferred to start their utterances with the animate entity regardless of position, suggesting that effects of animacy in adults have been overwritten by effects of literacy.

## Introduction

Whether something is alive, and potentially a social partner or a deathly threat, or whether something is merely a thing makes a huge difference. Paying attention to animacy thus seems critical for our survival and possibly reflects our evolutionary adaptivity [1]. Even young infants are already sensitive to certain properties of animacy. For instance, two-month-old infants can distinguish between animate and inanimate entities based on featural cues (e.g., faces, contour) and dynamic cues such as autonomous and contingent motion (for review, see

at https://osf.io/cz9ku/?view_only=
797d95a6c9684a9aa6c0b954a1e3276d.

**Funding:** This research has been funded by the
German Research Foundation (Deutsche
Forschungsgemeinschaft, DFG) - Project-ID
281511265 - as part of the CRC 1252 "Prominence
in Language" in the project B06 at the University of
Cologne. The funding was awarded to MP. The
funders had no role in study design, data collection
and analysis, decision to publish, or preparation of
the manuscript.

**Competing interests:** The authors have declared
that no competing interests exist.

[2]). Furthermore, infants attend differently to the interaction between two human actors compared to an interaction between a human actor and an inanimate object [3]. Animate entities also immediately attract our visual attention [1, 4]. For instance, when presented with two natural scenes, one including an animal, adults initiate a saccade to the scene with the animal in as little as 120 milliseconds [5]. As these findings illustrate, recognizing animacy is of vital importance for human beings.

The central role of animacy is also reflected in the structure of human language [6, 7]. Across languages, animate entities are typically placed before inanimate entities in a sentence, an observation known as Animate(d) First Principle [6, 8]. This preference is also supported by psycholinguistic research. For instance, in a sentence-recall task, Greek speakers tended to reverse the order of the sentential subject and object depending on the referents' animacy status [9]. When the subject was inanimate (e.g., *stone*) and the object animate (e.g., *man*), participants were more likely to misremember Subject-Verb-Object sentences (such as *The stone hit the man*) as Object-Verb-Subject sentences, underlining speakers' preferences to place animate before inanimate entities [9, 10]. Animacy also influences grammatical role assignment such that animate referents are more likely realized as the subject of an utterance [11]. For instance, when English speakers were asked to reproduce sentences involving an animate and an inanimate entity (e.g., *The music soothed the child*), participants were more likely to erroneously remember animate referents as subjects, even if this resulted in the production of a passive construction (e.g., *The child was soothed by the music*) [11]. Speakers are thus inclined to deviate from a prototypical syntactic structure when the patient outranks the agent in terms of animacy, as shown in numerous languages such as Spanish [12], Dutch [13], German [14], Japanese [10], and Tzeltal [15]. One explanation for these tendencies is that animate entities are more prominent as opposed to inanimate entities [16, 17], making them more accessible at a conceptual level and hence an optimal starting point for a sentence [18, 19].

Crucially, effects of animacy are not only present in adult speakers but they are operative from early on in language acquisition [20–24]. For instance, when children were taught passive constructions with animate patients such as *The baby is being picked up by the girl*, they were better able to produce and comprehend passives than children who were taught passives with inanimate patients such as *The flower is being picked up by the girl* [22, 23]. Similarly, child learners of Catalan were more likely to change the canonical Subject-Verb-Object (SVO) word order of a sentence to Object-Verb-Subject (OVS) when the patient was animate than when it was inanimate [24]. Taken together, these findings suggest that the animacy status of a referent plays an important role for language comprehension and production across the lifespan.

However, despite its pervasive effects at the sentence level, mixed results have been obtained for the role of animacy at the phrase level (i.e., regarding single constituents of a sentence such as conjoined noun phrases). For instance, when speakers of Japanese were asked to remember sentences like *The boat and the fisherman were moving*, they did not reverse the order of nouns when the inanimate noun preceded the animate noun [10]. Likewise, speakers of English showed no effect of animacy for coordinated noun phrases within sentences [11]. By contrast, participants in this latter study were more likely to produce the animate object first when they had to recall two nouns presented in isolation, outside of a sentence context (e.g., participants were more likely to produce *guest and piano* than *piano and guest*). Similarly, German-speaking adults and preschool children showed a preference to name the animate entity first when presented with two pictures in a picture-naming task [25]. Thus, whereas some studies show that animacy influences speakers' linear ordering of conjoined noun phrases, others do not find effects of animacy at the phrase level.

What could be the reason for these discrepant findings? On the one hand, it has been argued that animacy plays a major role for grammatical role assignment (i.e., by assigning the

animate entity the subject role) but that it should be irrelevant for linear ordering [11]. According to this view, animacy should not affect the sequence of conjoined noun phrases because animacy is exclusively linked to subjecthood. Note, however, that this view is incompatible with the observation that animacy can affect word order without changes in grammatical role assignment [9, 10]. For instance, speakers of Greek preferred to place animate entities in early word order positions, irrespective of their grammatical role, suggesting that animacy affects linear ordering [9]. Restricting animacy effects to grammatical role assignment also fails to explain why sometimes effects of animacy can be observed during the production of simple conjoined noun phrases [25]. On an alternative account, animacy could be just one among many features that affects accessibility and hence competes with other sources of influence. Indeed, there is evidence that the binomial ordering of nouns can be influenced by numerous factors, including differences in familiarity (familiar before unfamiliar), stress patterns (iambs before trochees), and frequency (more frequent before less frequent) [25–28]. It is therefore likely that multiple factors compete and potentially outrank animacy in terms of accessibility, rendering effects of animacy less influential when conjoined noun phrases are produced. In the present study, we directly test this possibility by focusing on one factor that seems particularly relevant for binomial ordering during picture naming, namely experience with the directionality of a writing system.

## Effects of reading and writing direction

There is evidence that experience with the directionality of a writing system (e.g., left-to-right vs. right-to-left) can affect constituent ordering during language production and comprehension. For instance, when Italian speakers (who are acquainted with a left-to-right script) were asked to draw depictions of simple transitive sentences like *The girl pushes the boy*, they preferred to place the agent on the left and the patient on the right. By contrast, the reverse was true for speakers of Arabic, who read and write from right to left [29, 30], but see [31]. This so-called spatial agency bias (SAB) was also confirmed by [32], who tested adult and child speakers of German, a language with a left-to-right writing system, and Hebrew, a language with a right-to-left writing system [32]. Whereas German-speaking adults showed a preference for placing agents on the left side, Hebrew-speaking adults preferably placed the agent to the right of the other arguments in an experiment where participants had to arrange depictions of the arguments presented in a sentence. By contrast, German and Hebrew preschool children displayed no spatial preference, suggesting that the SAB requires a sufficient amount of experience with a writing system in order to emerge. Further support for this assumption comes from a population of illiterate adults [33]: Speakers of Yucatec and Spanish who did not know how to read and write revealed no spatial preference when they were asked to draw events, suggesting that literacy can indeed influence how events are represented.

Experience with a writing system can also affect language production [34, 35]. For instance, in a picture description task, literate speakers of German were more likely to produce passive sentences when the patient of an action was located on the left rather than on the right, suggesting that the visual arrangement of event characters prompts speakers to assign the left-positioned character the subject role [34]. The directionality of a writing system also appears relevant at the phrase level. For instance, when English speakers were tested in a picture description task, they preferred to start with the leftmost depicted character when describing joined activities (e.g., *The cat and the dog vs. the dog and the cat—are growling at each other*) [36]. Converging evidence comes from a study by [37] who tested several cohorts of participants in a picture naming and recall task. Participants were either acquainted with a left-to-right script (Kannada), a right-to-left script (Arabic), or they were bidirectional readers of

Urdu and English [37]. Critically, the results showed that participants displayed spatial biases in accordance with their reading and writing habits. When acquainted with a left-to-right script, participants tended to name and recall the depicted items from left to right, whereas the opposite was true for readers acquainted with a right-to-left script. By contrast, no spatial biases were observed for illiterates, lending further support to the idea that literacy (i.e., the acquaintance with a particular script directionality) influences spatial biases during picture naming and recall (also see [38]).

As the abovementioned findings show, reading and writing habits can affect the ordering of constituents in a sentence and the ordering of nouns in conjoined noun phrases. Beyond these findings, there is some first evidence to suggest that the directionality of a writing system becomes highly influential, potentially even outranking other factors–such as animacy. For instance, when adult speakers of Dutch were asked to describe images of spatial arrangements by using relational terminology such as *between X and Y*, they displayed a systematic preference for mentioning left entities first, in line with a left-to-right reading direction [39]. By contrast, the animacy status of the entities did not exert a strong influence on Dutch speakers' responses, suggesting that a left-first preference is more influential than effects of animacy in adult speakers. This finding also fits with the observation that German-speaking adults–unlike children–displayed a systematic bias to start conjoined noun phrases with the left picture, in accordance with their reading and writing habits [25]. Critically, adults also displayed a much weaker effect of animacy than children, hinting at the possibility that for adults the preferred sequence of naming depends more on the directionality of the script than on animacy.

However, although previous findings seem to suggest that reading and writing habits become highly influential and might even outrank effects of animacy, a direct test of this proposal is currently missing. Up to now, cross-cultural studies on reading and writing directionalities have exclusively focused on spatial biases but did not consider effects of animacy [38]. By contrast, studies examining both animacy and spatial position have exclusively focused on participants who are acquainted with a left-to-right script (e.g., speakers of Dutch in [39]). This cultural bias raises the possibility that other factors such as the spatial arrangement of the stimuli have fostered a left-to-right preference, regardless of script directionality. Thus, in order to pin down the role of reading and writing habits for effects of animacy, it becomes vital to test participants who are familiar with different reading and writing directions (also see [40]). In the present study, we sought to address this issue by taking a cross-cultural and developmental approach.

## The current study

The goal of the current study was to answer the following research questions:

(RQ1) Does animacy influence the order of naming?

(RQ2) Does literacy influence the order of naming?

(RQ3) Does literacy override potential influences of animacy?

Concerning RQ3, the goal was to clarify whether experience with a writing system can outweigh effects of animacy when it comes to the linear ordering of information. If true, this could help to illuminate the role of animacy at the phrase level (which has led to inconsistent findings in the past, see e.g., [11, 25]). Furthermore, rather than studying literacy effects in isolation, the synopsis of different factors (i.e., literacy and animacy) provides novel insights into the mechanisms and scope of reading and writing effects: To what extent does experience with a writing system induce spatial biases and how does this interact with other factors such as animacy? In order to answer these questions, a cross-cultural perspective is indispensable. Thus, in addition to examining a group of literate German-speaking adults (henceforth 'German

group'), we also investigated a group of literate Arabic-speaking adults (henceforth 'Arabic group'), thereby overcoming sampling biases that frequently dominate the cognitive sciences [40, 41]. Additionally, we took a developmental approach by examining a group of pre-literate German-speaking preschool children.

Due to the Covid-19 pandemic, all participants were tested individually via a shared screen presentation using the videoconferencing tool Zoom. Participants were tested in a picture naming task in which an animate and an inanimate noun were depicted next to one another. The position of the animate noun varied systematically: On half of the trials, the animate entity was located on the left, whereas on the other half of the trials the animate entity was placed on the right. This way, animacy and position were juxtaposed in order to elucidate which of the factors exerts a stronger impact on participants' production of conjoined noun phrases.

We reasoned the following: If animacy was more important than reading and writing habits, participants should start their utterances with the animate entity regardless of its position (left vs. right). However, if indeed experience with a writing system becomes more decisive, then adult speakers should be affected by the position of an entity rather than by its animacy status. More specifically, adults' response patterns should be biased by their reading and writing habits. Whereas adult German speakers should be more likely to mention the animate entity first when it was located on the left than when it was located on the right, the reverse should hold true for speakers of Arabic. Finally, based on evidence that pre-literate children do not yet show systematic spatial biases due to script direction [42, 43], we expected children to be sensitive to the animacy contrast and thus more inclined to start their utterances with the animate entity, regardless of its position.

## Experiment 1: Binomial ordering of conjoined noun phrases in adults

### Methods

**Participants.** The German group consisted of 24 participants (16 female, 8 male, 1 left-handed, mean age: 29 years, range: 23–41 years). Three additional participants were excluded from the analyses, one because s/he was familiar with Hebrew (a right-to-left script) and two because they reported to have been following a strategic response behavior. All remaining participants were exclusively familiar with a left-to-right script. According to a language background questionnaire, all of the participants had learned English during high school. Furthermore, the majority of the participants had learned an additional language such as French or Spanish. Two participants reported to be Russian-German bilingual.

The Arabic group consisted of 25 participants (8 female, 17 male, 3 left-handed, mean age: 33 years, range: 19–58 years). Two additional participants had to be excluded: One because s/he received first schooling in Germany, and for the other, no data could be collected because the internet connection broke down. Three participants were Kurdish-Arabic bilinguals, one from birth, the other two learned Arabic in school at the age of six. Except for two (Lebanon, Palestine), participants were from Syria and were currently residing in Germany (mean years of residence: 7 years, range: 2–30 years). On average, the group of Arabic speakers received 11.5 years of schooling (min.: 7, max.: 12 years) in Arabic-speaking countries and all participants were literate in Modern Standard Arabic (right-to-left script). Due to foreign-language instructions at school in their home countries and due to German instructions after immigration to Germany, participants of the Arabic group were also exposed to left-to-right scripts (German/English). Thus, all participants of the Arabic group were familiar with two reading/writing systems (Arabic: right-to-left, and German/English: left-to-right) which they used in their daily life. In general, it should be noted that literate speakers of Arabic who learn to read

and write in Modern Standard Arabic (i.e., the written variety of Arabic), are always to some extent familiar with a left-to-right script as they are also taught to speak, read, and write in English. In our study, Arabic participants were additionally acquainted with reading and writing in German. To get an estimate of the extent to which Arabic participants were exposed to the German writing system, we asked them whether they use German as a language of communication at work or not.

Participants were informed about the study via personal communication and through advertisement by the University's International Office. Participation in this study was voluntary and could be terminated at any time during the experiment. All procedures were in accordance with the Declaration of Helsinki. In line with the rules of the German Research Foundation (DFG), the experiments with adult participants were exempt from an ethics vote. As specified by the DFG, psycholinguistic experiments that use non–invasive methods and that test healthy participants do not require a special ethics vote if the experiments do not pose a risk or physical/emotional burden to participants and as long as participants are debriefed about the study (see https://www.dfg.de/en/research_funding/faq/faq_humanities_social_science/index.html). Since we had previously obtained a positive ethics vote for the same experiment with children (see study 2), we could rule out any physical or emotional burden for adult participants. Participants received an email including information about the experiment and the data use as well as an invitation link to a Zoom session. Participants were informed that by taking part they agreed that their data was used for scientific purposes, thereby giving informed consent to participating in the study. Recruitment and testing of the German group took place from 8.4.2021 until 28.4.2021. Recruitment and testing of the Arabic group took place from 5.12.2020 until 26.4.2021.

**Materials and design.** We created 30 animate-inanimate noun pairs. Since we planned to use the same materials for the cohort of preschool children, all nouns selected either appeared in childLex [44] a corpus of children's books, or were nouns expected to be known by young children (e.g., *fox*). Animate nouns included only animals, no humans, and all inanimate nouns referred to stationary objects. The nouns were matched in terms of grammatical gender in German (masculine, feminine), syllable length (monosyllabic, bisyllabic) and frequency. There were 10 monosyllabic noun pairs and 10 bisyllabic noun pairs with masculine gender as well as 10 bisyllabic noun pairs with feminine gender. German also has neuter gender, however, masculine and feminine appear more frequently (see e.g., [45]). All bisyllabic nouns had a trochaic stress pattern, which corresponds to the predominant stress pattern in German [46]. All nouns were monomorphemic. Frequency data for adults were taken from SUBTLEX-DE [47, 48], a corpus based on German subtitles. Frequency data for children were taken from childLex for the youngest age group available, that is, six to eight years. A detailed overview of noun frequencies can be found in S1 File.

Depicted nouns that appeared together as a pair were matched in terms of their visual features by selecting pictures with similar size, detail, and shading/brightness; see Fig 1 for an example of our experimental item set. The nouns were depicted by non-colored drawings taken from the MultiPic database [49] or free databases, or photographs taken from the Bank of Standardized Stimuli BOSS; [50]) shaded in grey. We added 14 filler noun pairs consisting of trisyllabic nouns that were matched in terms of animacy (either both animate, or both inanimate).

The experimental design included the factor position (animate left vs. animate right) as within-subjects factor. The position of the animate entity was manipulated within-items as illustrated in Fig 1. Two presentation lists were created. In each list, the animate entity was located on the left for half of the trials, while for the other half, the animate entity was located on the right. The position of the animate entity was counterbalanced across the two lists (i.e.,

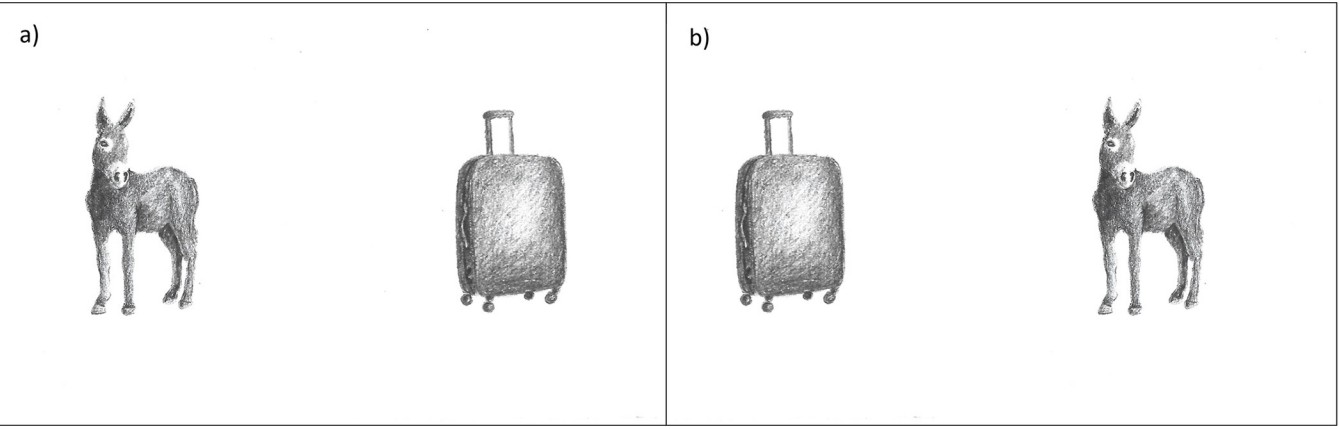

**Fig 1. Example item.** An example item of the picture-naming task. The position of the animate entity was varied systematically so it either appeared on the left (panel a) or on the right (panel b). Due to copyright reasons, the figure is not identical to the original image and is therefore for illustrative purposes only. The figure has been reproduced under a CC BY license, with permission from Eva Fielenbach, original copyright [2024].

mirror versions of the same item pairs were created such that a trial that displayed the animate entity on the left in list 1 would locate the animate entity on the right in list 2). Half of the participants were presented with list 1, the other half with list 2. Each participant saw each item pair only once. Overall, participants encountered 44 picture pairs that were presented one after another. The order of presentation was pseudorandomized so the same ordering (e.g., animate left) never occurred more than three times in a sequence and experimental items were followed by a filler after a maximum of four experimental items in a sequence. Trials were presented in the previously fixed pseudorandomized order.

**Procedure.**   Each Zoom session started with some general information about the procedure and a language background questionnaire. The Zoom session with the German group was led by a research assistant. The experimenter shared her screen with the participant and, first, noted down the participant's responses to the questions of the language background questionnaire. Then she continued with the picture-naming task. The task started with three example trials (e.g., *tree and dog*). The picture pairs of the example trials were arranged vertically so to not prime the participants by using a left-to-right or right-to-left direction, but only to familiarize them with the task. Two of the practice picture pairs were named from bottom to top, whereas one picture pair was labelled from top to bottom by the experimenter in order to illustrate the task. Next, participants were asked to name the experimental picture pairs (which were arranged horizontally, see e.g., Fig 1). Pictures remained on the screen until participants had finished their responses. During the task, the experimenter noted whether participants named the left or the right picture first on a coding sheet. After the experimental session, which altogether took around 10 to 15 minutes, participants were debriefed about the experimental goal.

The Zoom session with the Arabic group was led by one of the first authors (JS) and an assistant who was a native speaker of Arabic. The experimenter shared her screen with both the assistant and the participant in order to go through a language background questionnaire while writing down the participants' answers on a sheet before her. Afterwards, participants took part in a familiarization phase. While most of the communication before had been in German (if there were any clarification questions that could not be answered in German, the assistant translated the answer into Arabic), participants now had to read in the language they received schooling in (i.e., Modern Standard Arabic). During the familiarization phase, the

experimenter showed all of the pictures that were about to appear in the picture-naming task. Each picture appeared one after another with the corresponding Arabic name displayed below. This was done for two reasons: First, since we lack word frequency data for Arabic, this procedure ensured that the Arabic group was familiar with all of the experimental items. Second, the procedure was implemented to remind participants of the Arabic words. Note that all participants were currently residing in Germany and hence frequently exposed to the German language. We therefore sought to ensure that they did not encounter any difficulties in labeling the experimental items in Arabic. After the familiarization, participants were presented with the same three examples as the German group (e.g., *tree and dog*). Subsequently, participants were presented with the experimental items which they were asked to name in Arabic. For each trial, the assistant noted down the participant's responses on a coding sheet by ticking whether the left or the right picture was named first. After the experimental session, which altogether took around 15 to 20 minutes, participants were debriefed about the experimental aim.

## Results

All responses of the experimental trials were coded as left-first (1) or not (0). Likewise, we coded whether participants started their utterances with the animate entity or not (animate first = 1, animate second = 0). We included items for which participants produced another noun than expected (e.g., *Kröte* 'toad' instead of *Frosch* 'frog'). However, one trial had to be excluded from the German group because the participant was unsure how to name the picture. Twenty-eight trials (of 750 trials, 3.7%) had to be excluded for the Arabic group because participants could not name the pictures. Of these, 12 trials (3.2%) were excluded from the 'animate left' condition, whereas 16 trials (4.3%) were excluded from the 'animate right' condition. The maximum number of excluded trials for one Arabic speaker was four. This left us with 719 analyzable responses for the German group and 722 analyzable responses for the Arabic group.

**Descriptive results.** Experimental items: In the German group, 24 of 24 participants (100%) started their utterances with the left image on the majority of trials. By contrast, in the Arabic group, only 8 of 25 participants (32%) started their utterances with the left image on the majority of trials. Concerning animacy, 3 of 24 German speaking participants (12.5%) and 3 of 25 Arabic participants (12%) started their utterances with the animate image on the majority of trials.

While the German group produced 684 left-first responses (95%), the Arabic group produced 268 left-first responses (37%). Furthermore, there were five (right-handed) participants of the Arabic group who named all pictures from right to left and four (right-handed) speakers who named all pictures from left to right. All others switched at least once between left-first and right-first. Thus, we observed more variation for the Arabic group than for the German group. Concerning animacy, the German group produced 371 animate first responses (52%) and the Arabic group produced 339 animate first responses (47%).

Filler items (i.e., items that were matched for animacy): Whereas the German group produced 324 left-first responses (97%), the Arabic group produced 123 left-first responses (36%) for filler items. An overview of participants' responses can be found in Table 1.

**Statistical analyses.** For statistical analyses, we computed generalized linear mixed-effects models in R [51], using the *lme4* package [52]. To obtain *p*-values, we used the *lmerTest* package [53]. We started with the maximally specified model [54] and successively removed random slopes and intercepts when the maximal model failed to converge. The best-fitting converging model was selected based on the lowest AIC value [55]. For follow-up pairwise

**Table 1. Percentages of responses.**

| Position of the animate noun | German | | Arabic | |
|---|---|---|---|---|
| | Left first → | Right first ← | Left first → | Right first ← |
| **Animate left** | 97% (348) | 3% (12) | 34% (124) | 66% (239) |
| **Animate right** | 94% (336) | 6% (23) | 40% (144) | 60% (215) |
| **Filler items (animacy matched)** | 97% (324) | 3% (10) | 36% (123) | 64% (219) |

Percentage of left-first and right-first responses for left-positioned animate nouns and right-positioned animate nouns as well as for filler items in German and Arabic. Total numbers are given in parentheses.

comparisons, we used the *emmeans* package [56], which adjusts *p*-values using the Tukey method for multiple comparisons. To address issues of model convergence, we additionally calculated Bayesian generalized mixed-effects models, using the R package brms [57]. Weakly informative priors were used to fit the maximal models. For reasons of brevity, the results of the Bayesian models are reported in S1 File. We also report power analyses for the critical comparisons in S1 File. Power analyses were conducted a posteriori in order to assess whether sample sizes were appropriate. The data files, materials, and the R code for all models and power analyses are openly available on OSF at https://osf.io/x7n85/.

To get an estimate of how participants label pictures in the absence of conflicting animacy information, we first examined whether reading and writing habits affected the order of mention for pictures that were matched in terms of animacy (i.e., filler items). For this, we computed a mixed effects logistic regression model with 'language group' (German group coded as -0.5 vs. Arabic group coded as 0.5) as independent variable and order of mention (left first–yes vs. no) as dependent variable. The best-fitting converging model included random intercepts for subjects. The model revealed a significant effect of 'language group', Est. = -7.02, SE = 1.24, $z$ = -5.68, $p$ = < 0.001, demonstrating significant differences between the German group and the Arabic group in line with their reading and writing directionality. While the German group predominantly started their utterance with the left entity, the reverse was true for the Arabic group (also see Table 1).

To examine whether reading and writing habits affected the order of mention when position and animacy were juxtaposed, we then computed a mixed effects logistic regression model with 'language group' (German group vs. Arabic group) and 'position' (animate left vs. animate right) as independent variables and order of mention (animate first–yes vs. no) as dependent variable. We used deviation coding (0.5, -0.5) for both factors (for the factor 'language group', the Arabic group was coded as 0.5 and German group as -0.5; for the factor 'position', animate left was coded as 0.5 and animate right as -0.5). The best-fitting converging model included random intercepts for items and subjects. The model revealed a significant effect of 'position', Est. = 6.46, SE = 1.45, $z$ = 4.46, $p$ = < 0.001 but no significant intercept (i.e., no significant effect of 'animacy', Est. = -0.49, SE = 0.37, $z$ = -1.33, $p$ = .19). Furthermore, there was no significant effect of 'language group', Est. = 0.36, SE = 0.78, $z$ = 0.46, $p$ = 0.65, but a significant interaction between 'language group' and 'position', Est. = -16.73, SE = 2.99, $z$ = -5.60, $p$ < 0.001, demonstrating that the German and Arabic groups solved the task in different ways (also see Fig 2). A separate logistic regression model conducted for the German group revealed no significant intercept (i.e., no effect of 'animacy', Est. = 0.20, SE = 0.37, $z$ = 0.53, $p$ = 0.60) but a significant effect of 'position' (Est. = 7.67, SE = 0.57, $z$ = 13.56, $p$ < 0.001), in line with a left-to-right bias. Although a reverse tendency was observed for the Arabic group, a separate

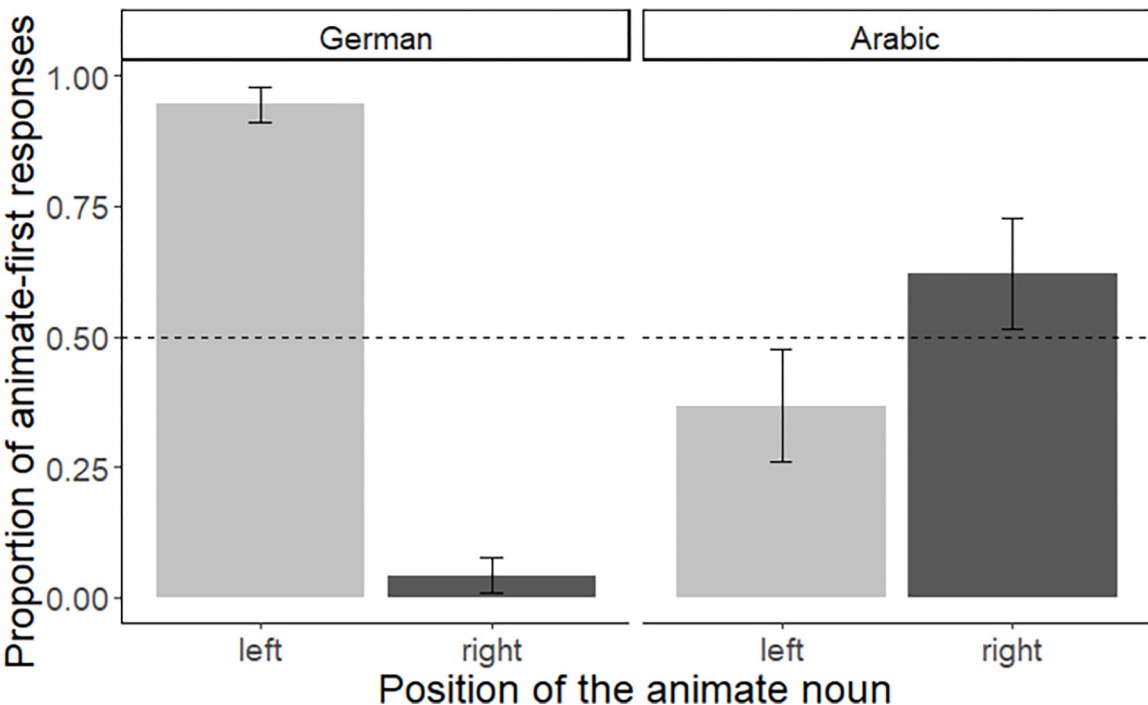

**Fig 2. Comparison between the German and Arabic group.** Mean proportion of animate-first responses for each of the two positions (i.e., animate entity located on the left vs. the right). Proportions are displayed for the German group (left panel) and the Arabic group (right panel). Error bars represent within-subject standard errors [58], calculated with the Rmisc package [59].

model for this cohort did not reveal a significant effect of 'position', Est. = -1.96, SE = 1.62, z = -1.21, $p$ = 0.23. Likewise, there was no significant intercept (i.e., no effect of 'animacy') for the Arabic group, Est. = -0.33, SE = 1.88, z = -1.74, $p$ = 0.08.

Given that the Arabic group was also familiar with the German left-to-right writing system, we sought to examine whether the amount of exposure to German influenced Arabic participants' responses. For this, we divided the data set according to whether Arabic participants used German as a language of communication at work or not (yes, n = 13 vs. no/irregularly, n = 12). In an exploratory approach, we first plotted the data and–based on this visual inspection–conducted a mixed effects logistic regression with the deviation-coded variables 'German at place of work' (yes coded as -0.5, no/mixed coded as 0.5) and 'position' (animate left coded as 0.5 vs. animate right coded as -0.5) on participants' animate-first responses (yes = 1, vs. no = 0). The final model included random intercepts for items. The model yielded no significant effect of 'German at work', Est. = 0.01, SE = 0.16, z = 0.07, $p$ = 0.94 but a significant effect of 'position', Est. = -1.16, SE = 0.16, z = 7.11, $p$ < 0.001, and, critically, a significant interaction between the factor 'German at work' and 'position', Est. = -2.24, SE = 0.33, z = -6.88, $p$ < 0.001 (see Fig 3). Post hoc comparisons revealed that Arabic speakers who did not use German at their place of work displayed a significant effect of 'position', in line with a right-to-left directionality, Est. = -2.28, SE = 0.25, z.ratio = -9.15, $p$ < 0.001. Thus, this sub-group of Arabic speakers was more likely to display a right-to-left bias (also see Fig 3, left panel). By contrast, Arabic participants who use German as a language of communication at work (and are thus more likely exposed to the German language and script) did not show a significant effect of 'position', Est. = -0.04, SE = 0.21, z.ratio = -0.18, $p$ = 0.86 (Fig 3, right panel).

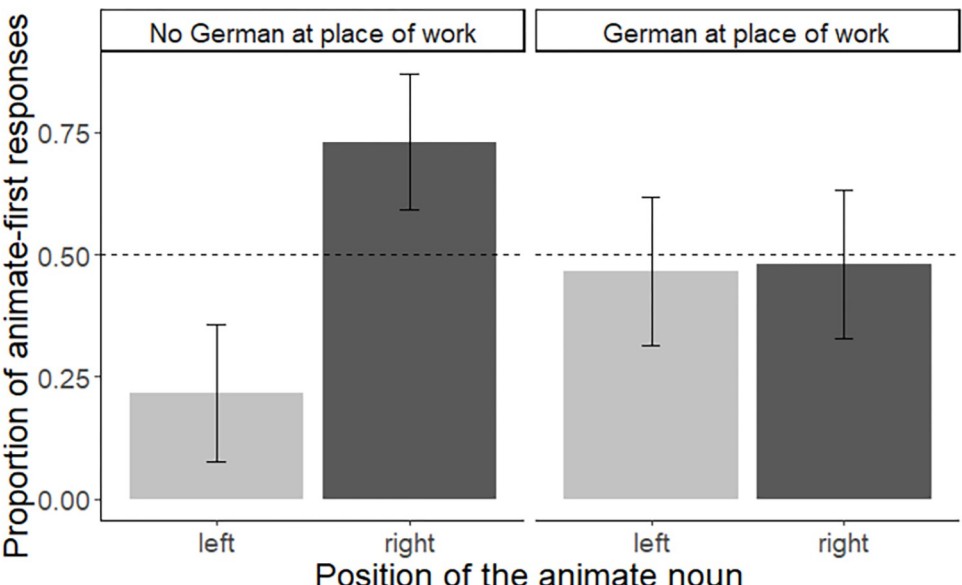

**Fig 3. Comparison between speakers of Arabic who use German at work vs. not**. Mean proportion of animate-first responses for each of the two positions (i.e., animate entity located on the left vs. the right). Proportions are displayed for speakers of Arabic who do not use German as a language of communication at work (left panel) and Arabic speakers who use German at work (right panel). Error bars represent within-subject standard errors [58], calculated with the Rmisc package [59].

## Discussion

For items that were matched in terms of animacy (i.e., filler items), German speakers displayed a left-to-right bias by mentioning the left entity first, whereas Arabic speakers displayed a right-to-left bias. These findings support previous observations that reading and writing habits can shape spatial biases during picture naming [37]. When juxtaposing animacy and position, our results show that animacy did not influence the ordering of nouns in conjoined noun phrases in adult speakers. Instead, picture naming was influenced by the position of an entity. With almost no exception, German speakers started with the left picture when presented with two pictures next to each other, even if the left picture showed an inanimate object and the right picture an animal. Whereas the German group was much more likely to mention the entity on the left first, irrespective of its animacy status, the Arabic group displayed a reversed tendency by preferentially starting with the entity depicted on the right. Taken together, our findings demonstrate that German and Arabic speakers–who are experienced with different writing directionalities–also differ in their spatial biases when naming pictures. However, the effect of position was attenuated in the Arabic group, most likely because this cohort was also familiar with the German writing system (i.e., following a left-to-right directionality). In support of this assumption, we found that Arabic participants who did not use German in a working context were more likely to follow a right-to-left strategy. By contrast, Arabic participants who also used German in their working environment (and who were likely more exposed to the German writing system), did not show the same bias. Note though that this finding has to be regarded as exploratory because it is based on a post hoc analysis. Nonetheless, the observed tendency is in line with previous observations by [60] who tested participants' spatial representation of events in a drawing and sentence-picture-matching task [60]. [60] found a reduced right-to-left bias for Arabic participants living in Italy at the time of testing as compared to Arabic speakers in their native country, suggesting that exposure to different writing

directionalities can mitigate spatial biases (also see [37] for similar observations). Taken together, our findings suggest that experience with a writing system can influence the way nouns are ordered in conjoined noun phrases. More specifically, we found that whether or not an animate entity was mentioned first critically depended on its location, demonstrating that for adult speakers reading and writing habits appear more influential for binomial ordering than animacy. However, while our findings hint at the possibility that animacy is no longer decisive for adult speakers, some caveats have to be taken into account. For one, it is assumed that animacy follows a hierarchy in which human beings outrank other forms of animacy (i.e., human > non-human animals > inanimate entities, see e.g. [61, 62]). Indeed, human vs. inanimate entities appear to form a particularly strong contrast in terms of animacy which is also reflected in the grammar of many languages, including Arabic (for effects of animacy on agreement options, see e.g., [63]). Consequently, a contrast between human vs. inanimate entities may have resulted in stronger effects of animacy than the one we observe here. While true, it should be noted that a contrast between animate (non-human) and inanimate entities still appears relevant for the linear ordering of conjoined noun phrases in German [25]. The same also seems to hold for Arabic [64]. That is, the animate entity is usually placed first when conjoined noun phrases are produced in Arabic (e.g. 'The camel and the desert have become a thing of the past'; [64]). However, while these findings confirm that animacy can affect linear ordering at the phrasal level, our results suggest that effects of animacy seem to be overwritten by reading and writing habits. In the next step, we sought to test this claim by examining preschool children who have not yet learned how to read and write.

## Experiment 2: Binomial ordering of conjoined noun phrases in German preschool children

### Methods

**Participants.**   The cohort of German speaking preschool children consisted of 24 participants (13 female, 11 male, 2 left-handed, mean age: 4;11 years, range: 3;8 to 5;11 years). According to a background questionnaire which was filled out by children's parents, all children were native speakers of German. Two of the children also had exposure to a second language (Spanish and Turkish, respectively), however, for all children German was the dominant language spoken at home. None of the children received speech therapy. Furthermore, all children were pre-literate and did not know how to read and write. Some of the preschool children had some very rudimentary writing skills (e.g., by being able to write their own name or a few words beyond their own name). All children were acquainted with picture books. The majority of children were exposed to reading activities (i.e., being read to by an adult) on a daily basis (n = 18), for the others this activity took place several times a week.

The study was conducted in accordance with the Declaration of Helsinki and approved by the Ethics Commission of Cologne University's Faculty of Medicine (approval number 16–134). Additional approval by the Ethics Commission was granted for the online collection and recording of children's data via Zoom. All caretakers gave informed written consent for their children to participate in the study.

**Materials and design.**   The same 30 animate-inanimate noun pairs as in Experiment 1 were used in Experiment 2, but we replaced three pictures by others that we expected to be easier to recognize by children (fox, whale, cookie). No fillers were included in the child experiment in order to minimize the duration of the task. Each child encountered 15 trials with the animate noun in left position and 15 trials with the animate noun in right position. The position of the animate noun was counterbalanced across two lists. In the child experiment, the task was embedded in a story. The children were shown a gust of wind and then were

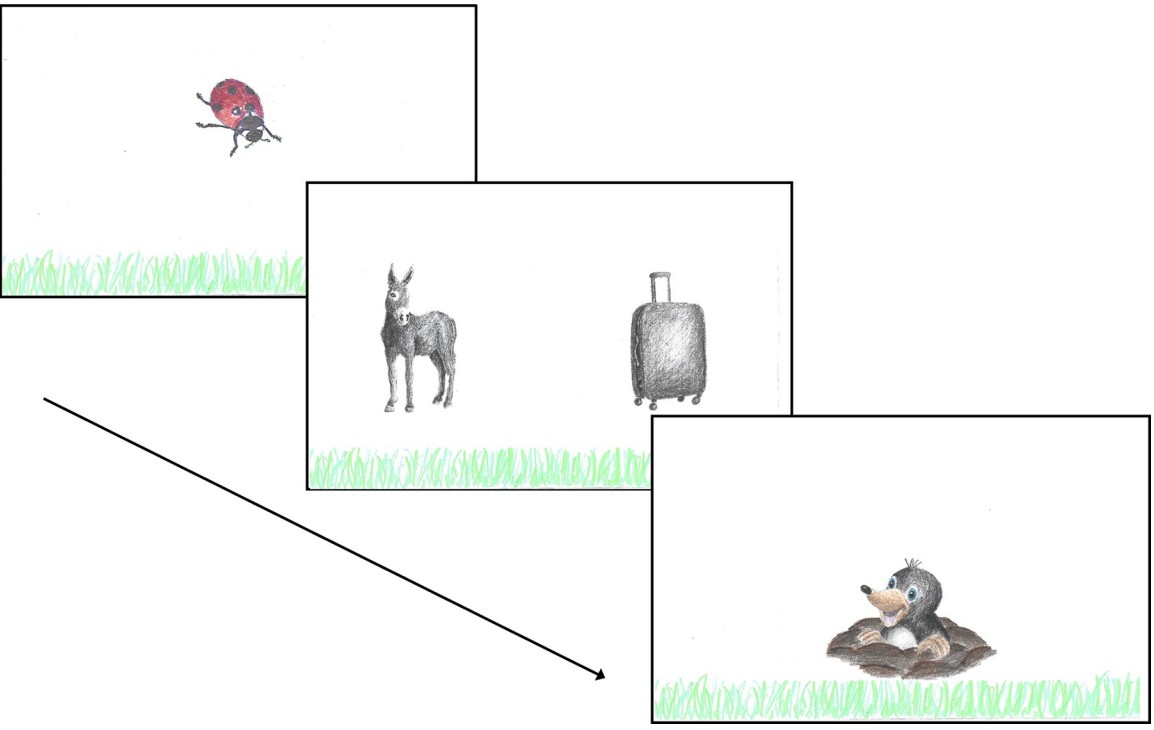

**Fig 4. Example trial (child experiment).** Example trial of the child experiment. A ladybug was included to guide children's attention towards the center of the screen before each trial. The children's task was to name the pictures for the little mole (The Mole, © Zdenek Miller/WDR mediagroup GmbH, note that due to copyright reasons, the figure is not identical to the original image and is therefore for illustrative purposes only). The figure has been reproduced under a CC BY license, with permission from Eva Fielenbach, original copyright [2024].

introduced to the little mole, a cartoon character well-known to German children (see Fig 4). Children were told that, because of the wind, the mole's pictures were now lost in the grass, and children were encouraged to help the little mole who cannot see very well by saying out loud which pictures they saw. Between trials, a ladybug was shown to guide children's attention towards the center of the screen. The children were told that there was no need to name the ladybug. To provide positive feedback, after 10 trials the little mole would jump up happily. In Fig 4, we provide an example trial of the child experiment.

**Procedure.** Before a testing session, the families were sent an information letter together with the consent form to be signed by both parents, a background questionnaire, and a short guide for using Zoom. Each test session was recorded, and the videos were deleted after children's responses had been transcribed and coded independently by two researchers. The session started with a quick camera check. Although we had initially planned to analyze children's eye gaze based on video recordings, children's viewing patterns could not be reliably determined. Therefore, we exclusively analyzed behavioral data (i.e., order of mention). During the session, the children were seated in front of a laptop or computer screen. The parents remained in the background. After a short introduction, the picture-naming task started with one practice item depicting a horse and a cow, and children were asked to say out loud what they saw. Once the children named both pictures in the practice trial, the experimenter would progress to the experimental items. Experimental items were presented on the screen until children had finished their verbal responses. An experimental session took around 10–15 minutes.

## Results

All responses of the experimental trials were coded as left-first (1) or not (0). Likewise, we coded whether children started their utterances with the animate entity or not (animate first = 1, animate second = 0). We included trials for which children produced another noun than expected (e.g., *Kröte* 'toad' instead of *Frosch* 'frog'), however, we excluded trials for which children produced an alternative noun with a different animacy status (e.g. *Pilz* 'mushroom' instead of *Qualle* 'jellyfish'). Likewise, we excluded trials when children were unable to name a picture or when parents prompted the child by providing a label. These criteria led to the exclusion of 84 trials (12%). Of these, 39 trials were excluded from the 'animate left' condition (10.8%) and 45 trials were excluded from the 'animate right' condition (12.5%). In total, this left us with 636 analyzable responses for the cohort of preschool children. Note that some of the children did not produce the conjunction 'and' while describing the depicted noun pairs. Hence, in the present study, the term 'conjoined noun phrase' can be conceived of as an umbrella term, encompassing constructions with and without the conjunction.

## Descriptive results

Eleven of 24 children (46%) started their utterances with the left image on the majority of trials. In total, preschool children produced 340 left-first responses (54%). Regarding animacy, 17 of 24 children (71%) started their utterances with the animate image on the majority of trials. Overall, children produced 367 animate-first responses (58%).

## Statistical analyses

For statistical analyses, we computed a mixed effects logistic regression model with position (animate left vs. animate right) as independent variable and order of mention (animate first–yes vs. no) as dependent variable. We used deviation coding (0.5, -0.5) for the factor position (with animate left coded as 0.5, and animate right coded as -0.5). The best-fitting converging model included random intercepts and slopes for subjects and random intercepts for items. The model revealed a significant intercept (i.e., a significant effect of 'animacy', Est. = 0.41, SE = 0.16, z = 2.60, $p$ = 0.01) but no significant influence of position, Est. = 0.40, SE = 0.48, z = 0.83, $p$ = 0.41 (also see Fig 5).

While all children were pre-literate and did not know how to read and write, we sought to examine whether some very basic writing experience was still sufficient to introduce spatial biases in children. To this end, we split the data file according to whether children already knew how to write a few words beyond their own name (n = 12) or not (n = 12) based on their parents' reports. We then computed a mixed effects logistic regression model for the deviation-coded factors 'basic writing' (yes coded as -0.5 vs. no coded as 0.5) and 'position' (animate left coded as 0.5 vs. animate right coded as -0.5) as independent variables and order of mention (animate first–yes vs. no) as dependent variable. The best-fitting converging model included random intercepts for items and subjects. Like before, the model revealed a significant intercept (i.e., a significant effect of 'animacy', Est. = 0.34, SE = 0.15, z = 2.35, $p$ = 0.02) but no significant influence of 'position', Est. = 0.30, SE = 0.17, z = 1.77, $p$ = 0.08. Critically, the model did not reveal a significant effect of 'basic writing', Est. = 0.04, SE = 0.23, z = 0.15, $p$ = 0.88, and no significant interaction between 'basic writing' and 'position', Est. = 0.01, SE = 0.34, z = 0.02, $p$ = 0.99. Although this finding is based on a post hoc analysis and hence to be considered exploratory, it suggests that some very basic writing experience did not affect children's spatial biases. To further address this question, we also calculated the correlation between children's age (in months) and their left-first responses (for illustration, also see S1 File). Age did not significantly correlate with children's left-first responses, $R$ = .29, $p$ = .17, suggesting again that

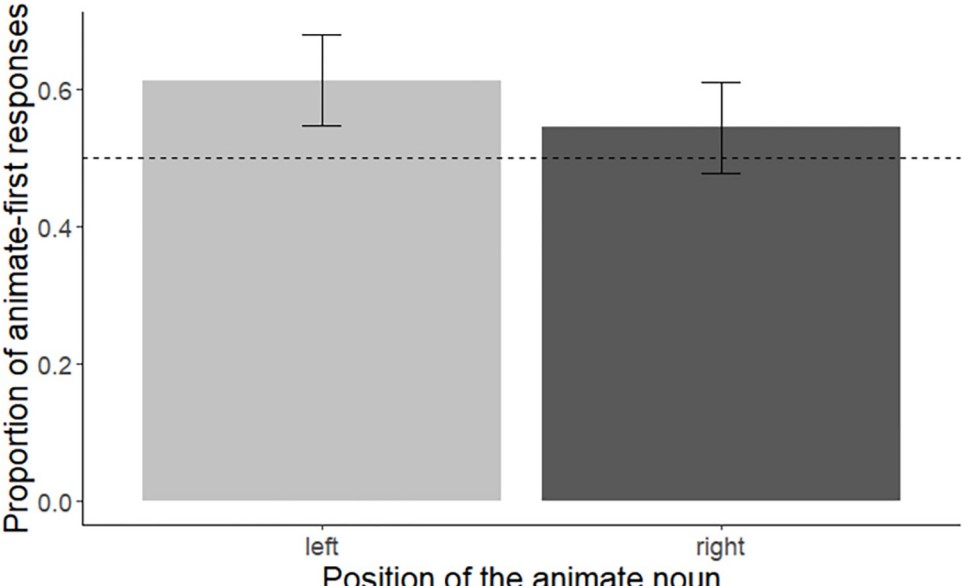

**Fig 5. Children's production of conjoined noun phrases.** Mean proportion of German children's animate-first responses for each of the two positions (i.e., animate entity located on the left vs. the right). Error bars represent within-subject standard errors [58], calculated with the Rmisc package [59].

more substantial exposure to a script is necessary in order to introduce consistent spatial biases in children.

Taken together, our results suggest that for children who do not yet know how to read and write, the animacy status of an entity is (still) important and not yet systematically affected by its position.

## Comparison of German children and adults

To examine whether binomial ordering differed for German adults and children, we computed a mixed effects logistic regression model with deviation coded factors 'group' (German adults coded as 0.5 vs. German children coded as -0.5) and 'position' (animate left coded as 0.5 vs. animate right coded as -0.5) as independent variables and order of mention (animate first–yes vs. no) as dependent variable. The best-fitting converging model included random intercepts for items and subjects. The model revealed a significant effect of 'position', Est. = 3.74, SE = 0.26, $z = 14.55$, $p < 0.001$ but no significant intercept (i.e., no effect of 'animacy', Est. = -0.30, SE = 0.19, $z = 1.61$, $p = .11$). There was no significant effect of 'group', Est. = -0.12, SE = 0.35, $z = -0.24$, $p = 0.73$, but there was a significant interaction between 'group' and 'position', Est. = 6.86, SE = 0.51, $z = 13.43$, $p < 0.001$, demonstrating that German adults and children solved the task in different ways. Unlike for German adults, who were much more likely to mention the animate entity first when it was located on the left rather than on the right, Est. = 7.18, SE = 0.48, z.ratio = 14.84, $p < 0.001$, the difference was not significant for children, Est. = 0.31, SE = 0.17, z.ratio = 1.83, $p = .07$.

## Discussion

Our results show that animacy influences children's binomial ordering of nouns. Children tended to mention the animate entity first–regardless of its position, suggesting that the conceptual accessibility of animates affects children's ordering preferences (in line with [25]).

Since our items were controlled for frequency, linguistic factors such as gender, morphological complexity and word stress, as well as for visual properties (i.e., image size, contrast, color), it appears unlikely that alternative factors underlie the results. However, it has to be noted that the effect of animacy was not particularly strong (only around 60% of children's responses were animate first). Again, this could be due to the fact that we did not select the strongest contrast in terms of animacy as we compared inanimate entities and animals instead of human beings. A central reason for choosing a distinction between animals (rather than human beings) vs. inanimate entities was the feasibility for children. That is, while most animals are well known to children and even 3-year-olds are already able to produce adequate labels, descriptions for humans are very limited if one does not take into account other specifications such as job descriptions (e.g. cook or fisher) which require the depictions of additional features (e.g., a special hat, a fishing rod, etc.). Depicting such features would also make it hard to ensure that animate and inanimate entities can be matched in terms of visual complexity, an important goal of the present study. Another potentially compromising factor could be that in books and movies, children often encounter inanimate cartoon characters that bear animacy features (e.g., talking cars, gesturing carpets, etc.), hence blurring the line between animate and inanimate categories. While in our study, we took care to use pictures of inanimate entities devoid of any animacy features (such as facial features, or autonomous movement), it is still possible that exposure to non-living yet 'animate-like' cartoon characters has mitigated the animacy effect in our study. Despite these caveats, our findings show that children were still sensitive to the animacy status of an entity. These findings support the idea that children's ordering of conjoined noun phrases draws on the conceptual distinction between animate and inanimate entities. By contrast, the position of an entity was not decisive for children's sequence of producing conjoined noun phrases, at least not when animacy was manipulated concurrently. This was true although all of the children in our sample were regularly exposed to joined book reading activities at their homes, and although half of the children had some very rudimentary experience with writing. While previous research has shown that such activities sometimes suffice to introduce spatial biases in preschool children–especially with regard to the so-called 'mental number line' in which smaller numbers are associated with the left and larger numbers with the right side of space [65–68], others find that directional biases are not yet present at a preschool age [32, 43]. In line with these latter findings, our results suggest that some basic exposure to reading is not sufficient but rather that more substantial experience with reading and writing is necessary for introducing spatial biases during picture naming. In sum, when presented with competing information, preschool children seem to base their ordering of conjoined noun phrases on animacy rather than on spatial position. This strategy contrasts with that of German adults, who order entities based on their position in a visual display.

## General discussion

In the present study, we sought to elucidate the impact of two different factors on speakers' production of conjoined noun phrases: the animacy status of an entity as well as its relative position in (horizontal) space. We find that preschool children who are not yet able to read and write, appear to be influenced by the animacy status of an entity but not (yet) by its position. At the same time, however, we observe that animacy is no longer decisive for adults' ordering of conjoined noun phrases, especially when spatial information is manipulated concurrently. Our findings therefore point to a developmental shift concerning children's and adults' ordering preferences–akin to what has previously been observed for influences of information structure [69]. For instance, when children and adults were asked to name two

different objects one of which they had encountered before, adults preferred to mention old/or given information first, whereas children displayed a novelty bias, mentioning a new entity before old information [69, 70]. Thus, children and adults show contrastive ordering preferences–a finding that also pertains to our results. However, it has to be acknowledged that our conclusions are exclusively based on a sample of German children. Ideally, we would have also collected data from preliterate Arabic children in order to test if the same developmental shift holds true cross-culturally. However, due to the special conditions under which this research had to be carried out (online testing during a pandemic), this was not feasible. While future studies should address this point, for now, our results suggest that children's linear ordering appears to be affected by animacy but not yet by reading and writing habits. Unlike children, adult participants who are acquainted with different script directionalities differ in how they order conjoined noun phrases, demonstrating an influence of reading and writing habits. Crucially, our findings further suggest that exposure to different writing systems appears to be more influential for adults' ordering of information than the animacy status of an entity.

However, do these findings mean that animacy is no longer relevant for adult speakers? While our results suggest that animacy may be more important for children than for adults, it should be acknowledged that animacy continues to be important for adult sentence production. For instance, when describing depictions of transitive events, speakers produce more passives when the patient is animate compared to inanimate patients [14, 34]. Furthermore, since the present study was conducted online, we exclusively focused on participants' order of mention rather than incorporating other measures such as eye movements or speech onset latencies. It is therefore possible that more subtle effects of animacy were not detected in the current set-up. While future studies may want to investigate this in more detail, our results show that the impact of animacy on the ordering of nouns seems to decline as a function of literacy development.

Our findings thus add to the body of research attesting a foundational role for reading and writing directions. Previous work suggests that experience with the directionality of a writing system influences how numbers are processed [71, 72], items are stored in memory [73], and even how pictures are judged in terms of their aesthetics [35, 74, 75]. In line with these findings, our results suggest that reading and writing directionality also plays a central role for language production. Beyond a spatial agency bias where reading and writing habits modulate how speakers map the most accessible (or prominent) referent, namely the agent, to a particular location in space, our results show that effects of reading and writing habits seem even more profound as they also affect the simple ordering of two otherwise unrelated entities. We find that the spatial arrangement of elements becomes decisive and–especially for speakers of German who are encultured to read and write from left to right–this habit completely determines the order in which elements are mentioned. Thus, the German group almost exclusively started to mention the left of the two images, regardless of animacy. In turn, this implies that whether or not German speakers first mentioned the animate entity was almost completely determined by its location. By contrast, a reverse tendency was observed for speakers of Arabic who have been exposed to a right-to-left script. However, in line with other findings [60], we did not observe a complete reversal in the Arabic group which is likely explained by the fact that Arabic speakers were also familiar with a left-to-right writing system. In support of this assumption, a right-to-left mapping was more pronounced for Arabic participants who were less frequently exposed to the German writing system since they do not use German in their working environment. While these findings suggest that spatial biases can be mitigated once people become literate in different writing directionalities, more research is needed to investigate the scope and mechanisms of these effects. How much experience with an additional writing system is necessary in order to shift spatial biases? While some research seems to imply

that spatial biases can be modulated quite rapidly even by short term training (e.g. by mirror-writing for a brief duration, [76]), other findings suggest that spatial biases need more time and input to emerge [37]. To shed more light on these questions, it becomes necessary to directly quantify the degree of exposure to different writing systems (see e.g., [77] for the same argument). We addressed this issue by asking Arabic speakers about their knowledge of German, specifically whether or not our Arabic participants used German in a working environment. However, more detailed measures may be needed in order to better quantify participants' exposure and degree of bidirectional reading. Ideally, one would also include participants who are exclusively literal in a right-to-left script. However, most of the cross-cultural work (including ours) is based on participants who have to be classified as 'bidirectional' readers. Yet, it should be noted that although Arabic speakers in our study were familiar with a left-to-right script, they still showed spatial biases which corresponded to the reading and writing system they were first acquainted with (i.e. right-to-left in Arabic), suggesting that spatial biases seem to persist to some degree even if participants are exposed to alternative reading and writing directionalities.

However, could other differences be the reason for the divergent behavior of the Arabic and the adult German group? As was shown previously, it is not only the exposure to a writing system but also typological properties of a language that determine the strength of spatial biases in language production and comprehension. For instance, whereas the spatial agency bias (SAB) appears particularly pronounced in languages with a strict word order, this bias is less strong for languages that allow for more flexibility in word order [78]. Crucially, however, these typological differences are unlikely to explain the differences we observe in our study. For one, both German and Arabic are rather flexible in terms of word order (see e.g. [79] for German, and [80] for Arabic). More importantly, we investigated a structure where differences in word order and other typological differences should be irrelevant–namely simple conjoined noun phrases. Although Arabic differs from German in terms of agreement options (in German, the verb has to agree with both conjuncts whereas in Arabic it is possible for the verb to exclusively agree with just one of the conjuncts, e.g. [81], this should also be negligible for our study as we did not ask participants to produce complete sentences including verbs but only conjoined noun phrases. Thus, while we cannot rule out that reasons other than literacy have contributed to the observed differences between our adult German and Arabic participants, cross-cultural differences in reading and writing directionalities appear to be the most likely and parsimonious explanation for our findings.

Our findings help reconcile previously inconsistent findings regarding effects of animacy on the production of conjoined noun phrases. We find that animacy can affect binomial ordering in children, thus replicating previous findings [25]. However, we also find that animacy by itself is no longer decisive for adults (thus corroborating findings by [11]), suggesting that other factors such as the position of an entity come to outweigh effects of animacy. More broadly speaking, our findings imply that there is competition between different factors regarding the accessibility and hence the ordering of noun phrases. While in some contexts animacy appears to be important, its effects 'vanishes' once other factors become more 'prominent'. This proposal is in line with other observations that the ordering of binomials can be influenced by multiple factors, including differences in familiarity (familiar before unfamiliar) or frequency (more frequent before less frequent) (e.g., [28]). Here we demonstrate that experience with a writing system constitutes yet another factor that can influence the accessibility of an entity, thereby shaping the binomial ordering of conjoined noun phrases. Most centrally, our results speak to the relative weight of animacy and reading and writing habits. Our cross-cultural and developmental findings suggest that–once acquired–reading and writing habits become influential to an extent that they overwrite alternative factors such as animacy. These

theoretical findings also come with practical implications: Given the wide-ranging influence of reading and writing habits, they have to be taken into account in experimental design. As our research shows, spatial biases arise even in verbal tasks that tap into patterns of language production, highlighting the need for taking effects of reading and writing into consideration.

## Conclusions

In literate adult individuals, the order of picture naming is influenced by reading and writing habits but not by the animacy of an entity. By contrast, the reverse is true for pre-literate children, suggesting this preference develops and that effects of animacy in adults may have been overwritten by effects of literacy. Taken together, our results provide new evidence that experience with a writing system is critical for how we order and process information, becoming influential to an extent that it can even outweigh effects of animacy.

## Supporting information

**S1 File. Additional analyses are provided.**
(DOCX)

## Acknowledgments

We would like to thank Barbara Zeyer and Esraa Naddaf for help with participant recruitment and testing, as well as for help with data coding. Thanks to all participants who participated in this study.

## Author Contributions

**Conceptualization:** Judith Schlenter, Martina Penke.

**Data curation:** Sarah Dolscheid, Judith Schlenter.

**Formal analysis:** Sarah Dolscheid.

**Funding acquisition:** Martina Penke.

**Methodology:** Sarah Dolscheid, Judith Schlenter, Martina Penke.

**Project administration:** Judith Schlenter, Martina Penke.

**Resources:** Martina Penke.

**Supervision:** Martina Penke.

**Visualization:** Sarah Dolscheid.

**Writing – original draft:** Sarah Dolscheid.

**Writing – review & editing:** Sarah Dolscheid, Judith Schlenter, Martina Penke.

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
