## [Decision Letter · Decision Letter 0]

2 Nov 2023

PONE-D-23-25451Literacy overrides effects of animacy: A picture-naming study with pre-literate German children and adult speakers of German and ArabicPLOS ONE

Dear Dr. Dolscheid,

Thank you for submitting your manuscript ID PONE-D-23-25451 entitled "Literacy overrides effects of animacy: A picture-naming study with pre-literate German children and adult speakers of German and Arabic" to Plos One. Your manuscript has been reviewed by two expert reviewers and by myself. The comments of the reviewers are included at the bottom of this letter. You will see that both reviewers are positive and provide feedback on how the paper can be improved; The Reviewer 2 considers that since no power analysis was conducted a-priori, you should present the power achieved for the critical comparisons. The Reviewer 1 and I showed concern over the same issue.

The number of different items to address are few, and thus they should be doable and will lead to a clearer and more robust paper. Please submit a new version of the manuscript by Dec 17 2023 11:59PM. If you will need more time than this to complete your revisions, please reply to this message or contact the journal office at plosone@plos.org. Please include the following items when submitting your revised manuscript:A rebuttal letter that responds to each point raised by the academic editor and reviewer(s). You should upload this letter as a separate file labeled 'Response to Reviewers'.A marked-up copy of your manuscript that highlights changes made to the original version. You should upload this as a separate file labeled 'Revised Manuscript with Track Changes'.An unmarked version of your revised paper without tracked changes. You should upload this as a separate file labeled 'Manuscript'.If applicable, we recommend that you deposit your laboratory protocols in protocols.io to enhance the reproducibility of your results. Protocols.io assigns your protocol its own identifier (DOI) so that it can be cited independently in the future. For instructions see: https://journals.plos.org/plosone/s/submission-guidelines#loc-laboratory-protocols. Additionally, PLOS ONE offers an option for publishing peer-reviewed Lab Protocol articles, which describe protocols hosted on protocols.io. Read more information on sharing protocols at https://plos.org/protocols?utm_medium=editorial-email&utm_source=authorletters&utm_campaign=protocols.

We look forward to receiving your revised manuscript.

Kind regards,

Montserrat Comesaña Vila

Academic Editor

PLOS ONE

3. We note that Figures 1 and 4 in your submission contain copyrighted images. All PLOS content is published under the Creative Commons Attribution License (CC BY 4.0), which means that the manuscript, images, and Supporting Information files will be freely available online, and any third party is permitted to access, download, copy, distribute, and use these materials in any way, even commercially, with proper attribution. For more information, see our copyright guidelines: http://journals.plos.org/plosone/s/licenses-and-copyright.

1. You may seek permission from the original copyright holder of Figures 1 and 4 to publish the content specifically under the CC BY 4.0 license.

Reviewers' comments:

Reviewer's Responses to Questions

**Comments to the Author**

1. Is the manuscript technically sound, and do the data support the conclusions?

Reviewer #1: Yes

Reviewer #2: Yes

2. Has the statistical analysis been performed appropriately and rigorously? 

Reviewer #1: Yes

Reviewer #2: Yes

3. Have the authors made all data underlying the findings in their manuscript fully available?

Reviewer #1: Yes

Reviewer #2: Yes

4. Is the manuscript presented in an intelligible fashion and written in standard English?

Reviewer #1: Yes

Reviewer #2: Yes

5. Review Comments to the Author

Reviewer #1: The MS PONE-D-23-25451 entitled “Literacy overrides effects of animacy: A picture-naming study with pre-literate German” presents the results of two experiments aimed to assess the influence of animacy on language processing, in particular on conjoined noun phrases. In the first experiment, data were collected from German and Arabic adult speakers. The results revealed a supremacy of reading and writing habits over any possible effect of animacy on language processing. In the second experiment, preschool children with no engrained habits of reading or writing were tested. Children “preferred to start their utterances with the animate entity regardless of position”, thus showing an animacy bias. Additional analyses were further conducted to explore the extent to which the amount of exposure to German influenced the Arabic adult participants’ responses. Exploratory analyses were also conducted regarding children’s writing experience. The conclusions were generally consistent with the predictions.

The MS is well-written, the experiments were designed carefully and efforts were made to ensure the procedure would be sensitive to the manipulation being implemented (e.g., some characteristics were equated between the animate and inanimate stimuli). The predictions are also clear in light of the presented literature and the discussion considers the related literature.

I applaud the authors for making their materials, data, and R codes available to the scientific community. I should note that the link provided in the “statistical analyses” section of the Manuscript leads to the “Materials” section on OSF. A clear note should also be made in the Manuscript on the fact that the materials are also made available.

I only have a few comments for the authors to consider.

At the end of the Introduction, the authors start by presenting their predictions and briefly describe the procedure. Then they have a section named ” The current study”. The first content could be integrated into such a section, thus avoiding some repetition of information that is occurring between these two parts; this would also make the reading more fluent and integrated.

Participants: some justification needs to be provided regarding the sample size in both experiments.

Experiment 1- “A separate logistic regression model conducted for the German group revealed no significant intercept (i.e., no effect of ‘animacy’, Est. = 0.20, SE = 0.37, z = 0.53, p = 0.60) but a significant effect of ‘position’ (Est. = 7.67, SE = 0.57, z = 13.56, p < 0.001), indicating that the German group was more likely to mention the animate entity first when it was located on the left than when it was located on the right.”- the conclusion of this sentence does not seem consistent with the data nor with the remaining conclusions and discussion.

Main Discussion: the ideas on page 33 regarding the effect of exposure of Arabic speakers to German are someone repeated. Please consider revising.

Another element that might be compromising a stronger effect of animacy in children derives from the fact that a lot of common objects are presented as animates in children’s programs. For example, a sponge becomes “Sponge Bob”, cars become animates (e.g., Lightning Mcqueen, Mater, and so on), a train becomes “Thomas the Tank Engine” and all his companions, Dora the Explorer's Backpack, and many others. Thus, the clear differentiation between animates and inanimates may become somewhat “unclear” with such experiences.

MINOR ASPECTS

• “mean age: 4;11 years, range: 3;8 to 5;11 years).” – the ; should be ,

• “statistical analyses” is used in Exp 1 whereas “statistical results” is used in Exp 2 to refer to the same type of content.

Reviewer #2: The study is straightforward, the method simple and the data are adequately analyzed.

The manuscript is well written but lengthy in view of its theoretical contribution. I would recommend shortening the text to help speed up the reading. I think that careful editing may potentially achieve a reduction of 20% of the length of the introduciton adn discussion.

One limitation of the present study is a choice of method affording one measure. More specifically, response latencies were not measured, which may have provided additional useful information (e.g., the comparison of RTs between the German group and those participants from the Arabic group who showed a left-to-right preference could provide evidence of a possible cognitive cost in the latter). This is not a major issue, however.

The authors do not appear to have based their sample on an a-priori power analysis. Low-powered studies incur a higher risk of false significant effects. The authors should justify their sample size and, since no power analysis was conducted a-priori, present the power achieved for the critical comparisons. For example, in Experiment 1, what was the power achieved given the sample size, Type I error of .05, and the effect size of the group x position of the animate noun interaction?

From a theoretical perspective, the study does not appear to make any important contribution. Are the results of relevance for theoretical models? What would be these implications? Can anything be said about the relative weight of animacy and reading/production habits? What could be the roles of reading vs language production in shaping the effects reported here?

6. PLOS authors have the option to publish the peer review history of their article (what does this mean?). If published, this will include your full peer review and any attached files.

Reviewer #1: No

Reviewer #2: No

---

## [Author Response · Author response to Decision Letter 0]

8 Jan 2024

Comments:

Editor:

You will see that both reviewers are positive and provide feedback on how the paper can be improved; The Reviewer 2 considers that since no power analysis was conducted a-priori, you should present the power achieved for the critical comparisons. The Reviewer 1 and I showed concern over the same issue.

We thank the editor and the reviewers for highlighting the merits of our study. We now provide power analyses and present the power achieved for the critical comparisons. Since both reviewers noted that the paper was too long and would benefit from editing, we present the results of the power analyses in the supplementary materials. The code and the detailed results can be found on OSF. Given the high power that was achieved for the critical effects, we can be quite confident that our study was not underpowered and that the selected sample size was appropriate.

Reviewer #1: 

The MS is well-written, the experiments were designed carefully and efforts were made to ensure the procedure would be sensitive to the manipulation being implemented (e.g., some characteristics were equated between the animate and inanimate stimuli). The predictions are also clear in light of the presented literature and the discussion considers the related literature.

I applaud the authors for making their materials, data, and R codes available to the scientific community. I should note that the link provided in the “statistical analyses” section of the Manuscript leads to the “Materials” section on OSF. A clear note should also be made in the Manuscript on the fact that the materials are also made available.

We thank the reviewer for the positive feedback and we now mention that the materials are also openly available (page 17).

I only have a few comments for the authors to consider.

At the end of the Introduction, the authors start by presenting their predictions and briefly describe the procedure. Then they have a section named ” The current study”. The first content could be integrated into such a section, thus avoiding some repetition of information that is occurring between these two parts; this would also make the reading more fluent and integrated.

Thanks, we have now edited the manuscript accordingly.

Participants: some justification needs to be provided regarding the sample size in both experiments.

We now provide (a-posteriori) power analyses to address this concern (also see response to the editor above). 

Experiment 1- “A separate logistic regression model conducted for the German group revealed no significant intercept (i.e., no effect of ‘animacy’, Est. = 0.20, SE = 0.37, z = 0.53, p = 0.60) but a significant effect of ‘position’ (Est. = 7.67, SE = 0.57, z = 13.56, p < 0.001), indicating that the German group was more likely to mention the animate entity first when it was located on the left than when it was located on the right.”

- the conclusion of this sentence does not seem consistent with the data nor with the remaining conclusions and discussion.

We now rephrased this statement by saying “in line with a left-to-right bias.”

Main Discussion: the ideas on page 33 regarding the effect of exposure of Arabic speakers to German are someone repeated. Please consider revising.

Thanks for pointing this out. We substantially revised and edited our manuscript in order to avoid redundancy. 

Another element that might be compromising a stronger effect of animacy in children derives from the fact that a lot of common objects are presented as animates in children’s programs. For example, a sponge becomes “Sponge Bob”, cars become animates (e.g., Lightning Mcqueen, Mater, and so on), a train becomes “Thomas the Tank Engine” and all his companions, Dora the Explorer's Backpack, and many others. Thus, the clear differentiation between animates and inanimates may become somewhat “unclear” with such experiences.

We thank the reviewer for this important remark. While in our study, we took care to use pictures of inanimate entities devoid of any animacy features (such as facial features, or autonomous movement), it is still possible that exposure to non-living yet ‘animate-like’ cartoon characters has mitigated the animacy effect in our study. We now acknowledge this possibility on page 28 of our manuscript. 

MINOR ASPECTS

• “mean age: 4;11 years, range: 3;8 to 5;11 years).” – the ; should be ,

We follow conventions in the language acquisition literature that list children’s ages in terms of year;months.

• “statistical analyses” is used in Exp 1 whereas “statistical results” is used in Exp 2 to refer to the same type of content.

Thanks, we have adjusted this accordingly. 

# Reviewer 2:

The study is straightforward, the method simple and the data are adequately analyzed.

The manuscript is well written but lengthy in view of its theoretical contribution. I would recommend shortening the text to help speed up the reading. I think that careful editing may potentially achieve a reduction of 20% of the length of the introduciton adn discussion.

Thanks for pointing this out. Based on this suggestion, we now substantially revised and shortened our manuscript in order to avoid redundancy.

One limitation of the present study is a choice of method affording one measure. More specifically, response latencies were not measured, which may have provided additional useful information (e.g., the comparison of RTs between the German group and those participants from the Arabic group who showed a left-to-right preference could provide evidence of a possible cognitive cost in the latter). This is not a major issue, however.

We agree that response latencies may have provided additional information, a limitation we have also pointed out in the original version of our manuscript (page 30). However, we also agree that this is not a major issue as the results are interpretable even in the absence of these additional measures. 

The authors do not appear to have based their sample on an a-priori power analysis. Low-powered studies incur a higher risk of false significant effects. The authors should justify their sample size and, since no power analysis was conducted a-priori, present the power achieved for the critical comparisons. For example, in Experiment 1, what was the power achieved given the sample size, Type I error of .05, and the effect size of the group x position of the animate noun interaction?

Thanks! We now provide a-posteriori power analyses of the power achieved (for the central cross-cultural analyses as well as for examining the power of the effects in children as opposed to adults). To ensure a better flow of reading, we have added the relevant information to the supplementary materials. The code is available at OSF. Overall, the achieved power is high and the selected sample size appears appropriate. 

We would also like to note that the sample size was determined on the basis of previous work investigating conjoined noun phrases in children (Franz et al., 2021). This study investigated 18 preschool children. Furthermore, we also conducted a lab experiment in which we tested adult speakers of German. In this pilot study (which we do not report in the current paper in order to ensure a better comparability since all of our other experiments were web-based), we found strong effects of German participants starting their utterances with the left-most entity, even when considering a small sample size of n = 20, hence serving as an anchor for determining the sample size in the current study. Based on this previous research, we then aimed at obtaining approximately the same number of participants for the other cohorts (Arabic group, German children).

From a theoretical perspective, the study does not appear to make any important contribution. Are the results of relevance for theoretical models? What would be these implications? Can anything be said about the relative weight of animacy and reading/production habits? What could be the roles of reading vs language production in shaping the effects reported here?

We thank the reviewer for challenging us on this point. In the revised version of our manuscript, we now highlight the theoretical contributions of our study as well as its methodological implications. Critically, although previous studies are suggesting that spatial biases introduced by reading and writing habits (i.e., script directionality) exert a stronger impact on linear ordering than animacy (e.g. Baltaretu et al., 2016), direct evidence of this claim is currently missing. This is mainly due to the fact that previous observations are exclusively based on participants who are familiar with a left-to-right script. It is therefore possible that all kinds of other factors (e.g. the spatial arrangement of entities or a very subtle manipulation of animacy, etc.) have contributed to the observed findings, regardless of reading and writing directionality. For this reason, our study is the first to directly test the notion that reading and writing habits become more influential than animacy. Hence, our results speak to the relative weight of animacy and reading and writing habits. Our cross-cultural and developmental findings suggest that – once acquired – reading and writing habits become influential to an extent that they overwrite alternative factors such as animacy. These theoretical findings also come with practical implications: Given the wide-ranging influence of reading and writing habits, they have to be taken into account in experimental design. As our research shows, biases arise even in verbal tasks that tap into patterns of language production, highlighting the need for taking effects of reading and writing into consideration.

---

## [Editor Report · Decision Letter 1]

30 Jan 2024

Literacy overrides effects of animacy: A picture-naming study with pre-literate German children and adult speakers of German and Arabic

PONE-D-23-25451R1

Dear Dr. Sarah Dolscheid,

We’re pleased to inform you that your manuscript has been judged scientifically suitable for publication and will be formally accepted for publication once it meets all outstanding technical requirements.

Kind regards,

Montserrat Comesaña

Academic Editor

PLOS ONE

---

## [Editor Report · Acceptance letter]

27 Mar 2024

PONE-D-23-25451R1 

PLOS ONE

Dear Dr. Dolscheid, 

I'm pleased to inform you that your manuscript has been deemed suitable for publication in PLOS ONE. Congratulations! Your manuscript is now being handed over to our production team.

Kind regards, 

on behalf of

Dr. Montserrat Comesaña Vila 

Academic Editor

PLOS ONE